# Intelligent Design of ZVS Single-Ended DC/AC Converter Based on Neural Network

**Nikolay Hinov** [1,*] and **Bogdan Gilev** [2]

1  Department of Power Electronics, Technical University of Sofia, 1000 Sofia, Bulgaria
2  Department of Mathematical Modeling and Numerical Methods, Technical University of Sofia, 1000 Sofia, Bulgaria
*  Correspondence: hinov@tu-sofia.bg; Tel.: +359-2965-2569

**Abstract:** This paper presents a model-based and neural network-based innovative design of single-ended transistor resonant DC/AC converters with zero voltage switching (ZVS). A characteristic of the proposed design method is that the determination of the circuit elements of the converter is performed with an automated procedure, as their values are determined by the output of a previously trained neural network. The use of the proposed method is justified in cases where there is no methodology for the design of the specific power electronic device, or such a methodology exists, but it is either too complex or based on a large number of assumptions. This is usually due to the increasing complexity of power circuits, their possible modes of operation, and the inevitable assumptions and limitations in the analyses and methodologies based on them. In this way, a natural combination of classic design methods and innovative processes is developed based on applied techniques for artificial intelligence.

**Keywords:** intelligent modeling and design; neural networks; single-ended DC/AC converters; zero voltage switching (ZVS); resonant converter

## 1. Introduction

Power electronics is an interdisciplinary field that is focused on the development of power electronic devices and electrical energy conversion systems. In this aspect, improving the design of these devices is an important task that has a direct impact on the efficient use of electrical energy. Usually, the design of electronic converters ends with optimization based on a certain objective function. Over time, different design methods have been established—classical methods based on analysis and design methodologies and innovative methods based on the application of mathematical modeling and software products.

In classical methods, the final result depends significantly on the human factor, which makes them a function of the experience and preparation of the designers. Due to the wide variety and wide application of power electronic devices, alternative design methods based on the application of artificial intelligence techniques have recently been sought. This is due to the development of computational mathematics and numerical methods, and to the successful implementation of information and communication technologies in scientific research and commercial development.

Artificial intelligence (AI) is one of the most remarkable and dynamically developing research fields in the last few decades. The goal of AI is to design and create systems with intelligent functions similar to biological learning, reactions, and interactions. AI has tremendous adaptive advantages and has been successfully applied in numerous industrial fields, including 2D and 3D image classification, speech recognition, mobile autonomy, computer vision, and more.

Thanks to the rapid development of big data science, sensors and sensor networks, Internet of Things (IoT), and computing and cloud technologies, a wide variety of data exists for power electronic systems in different phases of their life cycle. The growing volume of

data provides enormous opportunities and lays a solid foundation for the application of AI in power electronics. AI is able to use data to improve product competitiveness through global design optimization, intelligent control, performance evaluation, system aging prediction, and more. Due to the specific challenges and characteristics of electronic systems (for example, high tuning speed in control, high sensitivity in condition monitoring to detect aging, etc.) the implementation of AI in power electronics has its own characteristics that are different from other engineering fields.

In [1–3], methods are proposed for predicting the load of both single devices and power electronic systems in smart networks. In this way, an assessment was made of the range of parameter changes, as well as the possibilities of implementing relevant operating modes to cover energy needs. Artificial intelligence techniques have found widespread use in the synthesis of power electronic devices using a variety of applications [4–9]. In this aspect, various tools have been developed to support and assist with this process, with an aim to obtain robust control despite a wide change of both scheme parameters and disturbance effects. Another main direction for the application of artificial intelligence techniques in power electronics is assessing the reliability and improvement of the operation of power electronic devices and systems [10–12].

The main problems solved by the application of artificial intelligence are related to the modeling and simulation of various malfunctions, as well as to the prediction of accidents in individual devices and systems. A characteristic of this field of application is that we work with uncertainty and lack sufficient collected data on the operation of power devices and systems in different modes and working conditions. Artificial intelligence techniques make it possible to take into account aspects related to the operation and reliability of devices already at the initial stages of the overall design and prototyping processes. A natural development of these approaches is their implementation in systems for automated design of power electronic devices. A few specific cases of model-based design in various applications are presented in [13–19]. For this purpose, specialized expert systems have been developed and implemented [20].

On the other hand, one of the trends in the development of scientific research is the increasing use of neural networks and machine learning. For this purpose, various tools based on mathematical software have been developed. In [21–25], in-depth studies of the application of artificial intelligence techniques in the modeling, analysis, design, and operation of power electronic devices and systems are presented. It has been shown that the effectiveness of system autonomy and adaptability can be improved through the application of AI in the design and prototyping of power electronic devices and systems. In this way, devices with guaranteed indicators, self-learning, and self-adaptation capabilities useful for changing system input parameters, as well as tolerances of circuit elements and disturbances, have been realized.

AI applications for power electronics are categorized into the following aspects: modeling, design, control, and operation. AI methods typically include expert systems, fuzzy logic, heuristics and machine learning, deep learning with neural networks, and combined neural fuzzy systems.

In [26,27], the authors applied an approach for the design of DC-DC converters based on the use of neural networks. The aim of these studies was to prove the applicability of this new and innovative approach to the design of power electronic devices. The studies were carried out on basic schemes of DC-DC converters, such as Buck and Boost, because with them, the interpretation and verification of the obtained results are very quick. The main goal of formulating this design approach is to apply it to more complex power circuits, with more possible operating modes. In this way and in these cases, it would be possible to carry out a formalization, a mathematical description of the electromagnetic processes, and to apply an engineering methodology to automated design, accordingly.

The aim of the present work is, by applying techniques of artificial intelligence, such as neural networks, to realize a model-based design of a serial single-ended DC/AC converter with direct mode detection. In this way, the main advantages of artificial intelligence com-

pared to classical methods will be demonstrated, such as: the greater degree of automation and, accordingly, the speed of execution of the design process; weak dependence on the human factor; and the ability of algorithms to extract regularities from data and to self-learn and self-improve on this basis.

The manuscript is organized as follows: the first chapter introduces the importance of issues related to the design and prototyping of DC/AC converters; in the second chapter, the essence of the innovative approach to the design of power electronic devices, based on the use of neural networks, is presented; in the third chapter, this approach is demonstrated through the implementation of a specific calculation method for the design of a single-ended series resonant DC/AC converter; and the fourth chapter describes the main findings and conclusions, as well as guidelines for the future application and development of scientific research.

## 2. Materials and Methods

Depending on the type of resonant circuit, the single-ended transistor resonant DC/AC converter has two circuit types—series and parallel [28]. The difference between them is in the way of connecting the resonant capacitor and, accordingly, in the conditions necessary to ensure operation in ZVS mode. In this manuscript, the design of the series resonant DC/AC converter will be discussed, and based on this, our innovative approach to the design of power electronic devices will be formulated and presented.

### 2.1. Principle of Operation of the Single-Ended Resonant DC/AC Converter

Figure 1 shows a diagram of a series single-ended resonant DC/AC converter. The scheme consists of a load with an active-inductive nature (*L* and *R*), transistor T, capacitor *C*, and reverse diode D, which is necessary in cases where the transistor does not have a built-in one.

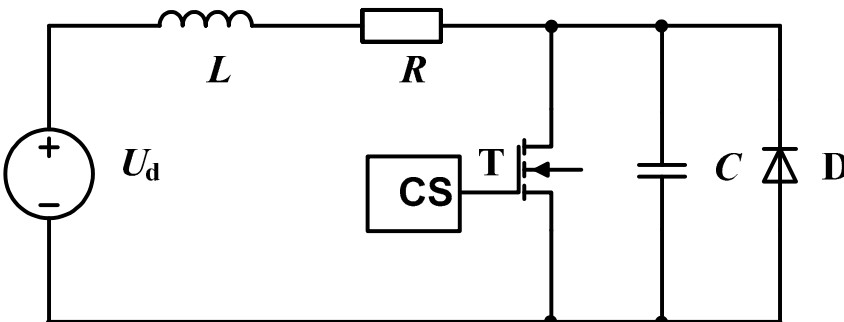

**Figure 1.** Single-ended series resonant DC/AC converter.

The determination of the basic ratios in the power circuit is made under the assumption of ideality of all building elements, and it is also assumed that the DC power source has zero internal resistance.

The action of the DC/AC converter is explained by the time diagrams of Figure 2. The operation of the power circuit, depending on whether the transistor reverse diode structure conducts or not, may be divided into two stages.

1-First stage: this is the interval that begins when a control pulse is applied to the transistor T. Initially, the reverse diode D turns on because the current through the inductance is in the reverse direction and the load circuit closes through it (for the interval from 0 to $t_D$, as shown in Figure 2). After the change in the direction of the current, the transistor T begins to conduct (for the interval from $t_D$ to $t_i$). A characteristic of this stage is that for the entire interval, an increasing current with a shape close to linear flows through the LR load, and the capacitor *C* has a voltage close to 0 (the voltage drop of the unblocked reverse diode D and transistor T).

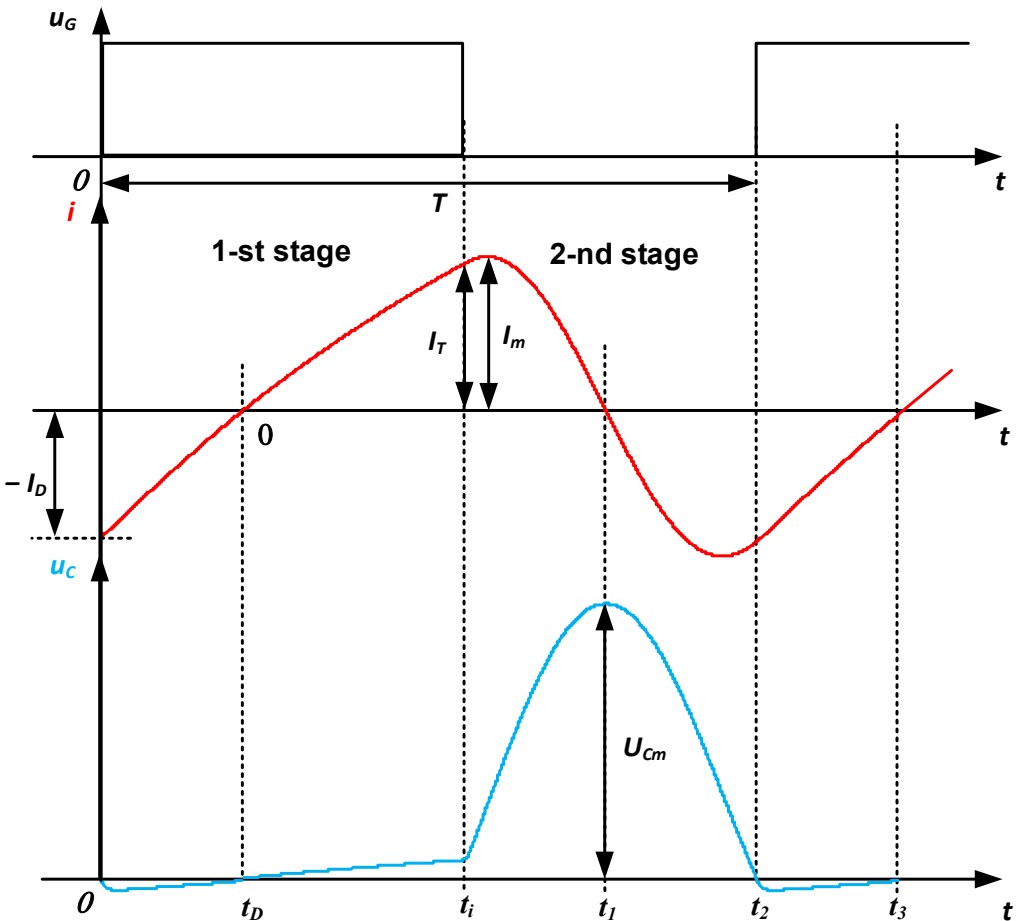

**Figure 2.** Timing diagrams describing the operation of a series resonant inverter. From top to bottom: control pulses, load current, and voltage on the capacitor and the transistor (on the reverse diode).

The second stage (from $t_i$ to $t_2$): it begins at the moment when the transistor T turns off. If resonance conditions are met in the series RLC circuit (and this is necessary for the normal operation of the power circuit), a sinusoidal current with a non-zero initial value $I_0$ begins to flow through the load (according to Figure 2). This initial value of current through the load is the value reached when the first stage of operation of the circuit has ended. Due to the presence of resonant processes, resonant capacitor $C$ is charged according to an oscillatory law to a voltage higher than the voltage of the DC power source. After that, a discharge process begins in the circuit, and when the voltage on the capacitor $C$ reaches the value of the input voltage $U_d$, the current through the inductance has a value close to the maximum, although with the opposite sign. Thus, the energy stored in inductance $L$ causes the further discharge of resonant capacitor $C$. When the capacitor voltage becomes equal to zero and tends to go negative, the reverse diode is turned on, which ends this interval of circuit operation, and the next period begins.

To achieve safe and reliable operation of the circuit and of the transistor with zero voltage switching (ZVS), at the moment of resetting the capacitor voltage (turning on the reverse diode D), it is necessary to provide control of transistor T. It will turn on when the current through the load reverses its direction and becomes equal to zero. When the transistor is turned on again, one period of circuit operation ends.

Considering the operation of the DC/AC converter, it is clear that it always works with zero initial conditions of the reactive elements (load current and capacitor voltage), which gives us precisely the reason to claim that the circuit works directly in an established mode. The adjustment of the output voltage and power in the scheme is carried out by changing the magnitude of the current at which the transistor T—$I_0$ is turned off.

### 2.2. Modeling of Operation of the Single-Ended Resonant DC/AC Converter

In the studied circuit, the state variables are *i* (the current through the inductance) and *u* (the voltage on the resonant capacitor). Normal operation of the circuit in soft-switching (ZVS) mode requires that the transistor be driven when the following conditions are simultaneously met for the current *i*: *i* increases and $i \leq I_0$.

Using Kirchhoff's laws, the following mathematical model is obtained. It describes the electromagnetic processes in the converter:

$$L\frac{di}{dt} = -R\,i - u_C + U_d$$
$$C\frac{du}{dt} = i \,.\, control(t) \tag{1}$$

where the switching function *control* defines the two states of the power circuit according to the two stages of its operation:

$$control(t) = \begin{cases} 1, & for \ \{i - \text{increases} \quad and \ \ i \leq I_0\} \\ 0, & for \quad \text{otherwise} \end{cases} \tag{2}$$

From a modeling perspective, the DC/AC converter is a partially linear system. It operates in an arbitrary number of steps, solving the resulting system of ordinary differential equations at each step. The forms of the state variables are learned such that for each step, the initial conditions are the values of the state variables at the end of the previous step, i.e., the system is solved by the stitching method (stich method).

The model thus defined was implemented in Matlab after substitution with the following values of circuit parameters: input voltage $U_d$ = 25 V, active resistance *R* = 1 Ω, capacitance *C* = 10 μF, and inductance *L* = 1 μH. The value of the current $I_0$ was chosen to be 15A. Figure 3 shows the obtained time diagrams of the current through the inductance and of the voltage on the capacitor. These circuit parameter values were obtained using the design methodology presented in [28].

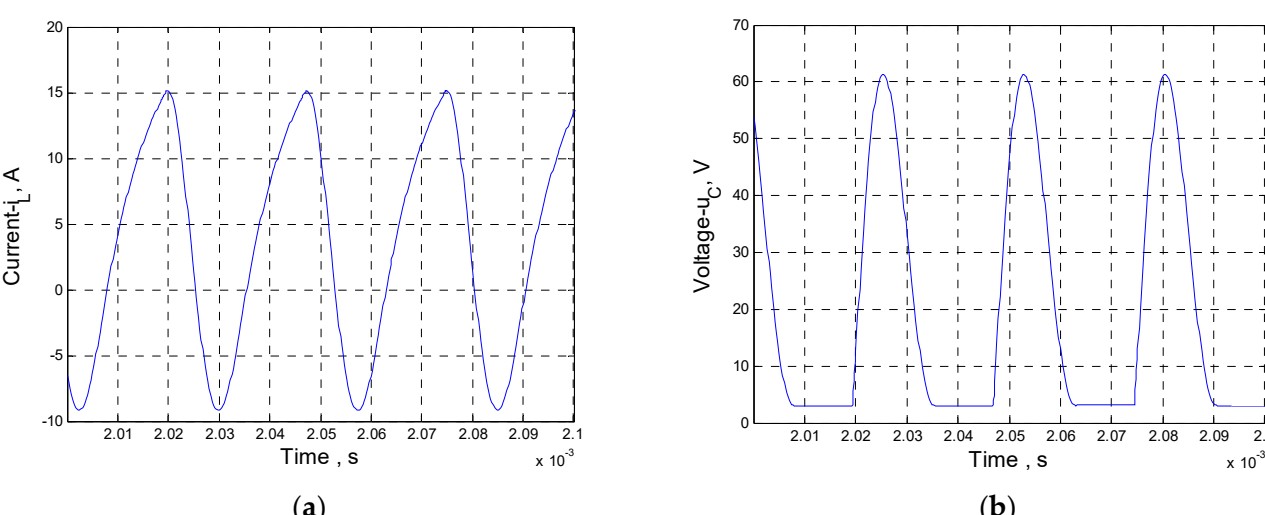

(**a**)  (**b**)

**Figure 3.** Timing diagrams of the DC/AC converter state variables: (**a**) current through the inductance; (**b**) voltage across the capacitor.

### 3. Results

In this section, the step-by-step design of a series single-transistor DC/AC converter with neural network will be presented in detail. The main tool for the creation and implementation of this procedure was the mathematical software Matlab. A strong argument for its use is that it is well-known to most engineers and designers and it is studied in most leading technical universities. In addition, it has a very large set of standard and advanced mathematical functions, which allows a significant volume of tasks and problems to be solved by people without a specific education in mathematics.

### 3.1. Formulation of the Task

For the design of the single-ended DC/AC converter, the forward neural network (NN) feed was used. In this aspect, for the successful application of the method, it was necessary to reduce the task to a static one. When changing some of the circuit parameters (*R*, *L* and *C*), the operating mode changes, and the dynamics of the behavior of the state variables *i* and *u* change accordingly over time. Based on the time diagrams of the current through the inductance *i* and the voltage across the resonant capacitor *u* (Figure 3), which had already been obtained through the model, we selected values suitable for defining the output of the neural network intended for design. These are, respectively, the mean values and maximum deviations of the state variables *i* and *u*, that is:

$$i_{aver} = \left( \sum_{k=1}^{n} i_k \right) / n, \ \Delta i = max(i_k) - \min(i_k) \tag{3}$$

$$u_{aver} = \left( \sum_{k=1}^{n} u_k \right) / n, \ \Delta u = max(u_k) - \min(u_k) \tag{4}$$

Of course, in order to eliminate the random fluctuations of the considered quantities, these four average values defined in this way were calculated sometimes after the start of the numerical experiments (to eliminate the transient process). In this case, the interval chosen was $t \in [2.10^{-3}, 5.10^{-3}]$.

Choosing $i_{aver}$, $\Delta i$, $u_{aver}$ and $\Delta u$ as the input for the network allowed us to change some of the values of the circuit elements *R*, *L*, and *C*, as well as the values of $i_{aver}$, $\Delta i$, $u_{aver}$ and $\Delta u$. Thus, the considered design task was reduced to a static one. This is what allowed the use of the forward NN for the automated design of the DC/AC converter. To visualize the design process, we called this network the inverse neural network model of the converter. If the design task had not been reduced to a static one, it would have required the use of recurrent neural networks in the role of an inverse neural network model. This would have significantly complicated the task, as it is not known whether it is possible to realize the modeling of the desired process with a recurrent neural network. On the other hand, the static task risked that the neural model would turn out to be inadequate for our applications (too simplistic). The results presented below show that, fortunately, this was not the case.

### 3.2. Preparing the Data for Training the Neural Network

The neural network training data were automatically generated with specially designed MATLAB source code, which was based on the mathematical model (1–2) and formulas (3) and (4). For the mathematical model, respectively, the input data were *R*, *L* and *C*, and the output data were $i_{aver}$, $\Delta i$, $u_{aver}$ and $\Delta u$, as shown in Figure 4.

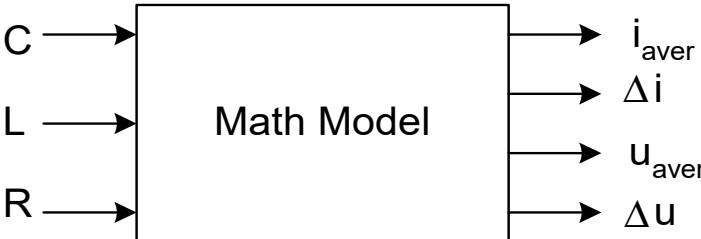

**Figure 4.** Input and output data of the mathematical model of the single-ended DC/AC converter.

In order to begin the input/output/model data generation process, we needed to determine the ranges of variation of *R*, *L*, and *C*. For the circuit in Figure 1 to work correctly, the parameters *R*, *L* and *C* must be selected such that:

- First, the condition for the resonance of the series circuit is satisfied: $R < 2\sqrt{\frac{L}{C}}$;

- Secondly, within a pseudo-period, the voltage on the capacitor u becomes negative and as a result, current flows through the reverse diode for some time.

In this aspect, some suitable ranges for changing $R$, $L$, and $C$, at which the above two conditions are met and which are close to the originally selected values of $R = 1\ \Omega$, $C = 10\ \mu F$, and $L = 1\ \mu H$, are as follows:

$$R \in [0.2;\ 1]\ \Omega,\ L \in [10;\ 18]\ \mu H \text{ and } C \in [0.6;\ 1.4]\ \mu F \tag{5}$$

On the other hand, it is quite possible to generate input/output data for the model (straight mesh) by supplying random values for the parameters $R$, $L$, and $C$ that fall within the specified limits. Due to the simpler program implementation, the associated improved speed, and the correspondingly lower requirements for the computing equipment, another approach was chosen here, where $R$, $L$ and $C$ were changed with a constant step, $\Delta R = 0.2\ \Omega$, $\Delta L = 2\ \mu H$ and $\Delta C = 0.2\ \mu F$ and were combined with each value of the other two parameters (three nested cycles). Thus, the $R$, $L$, and $C$ data shown in Figure 5 were initially generated.

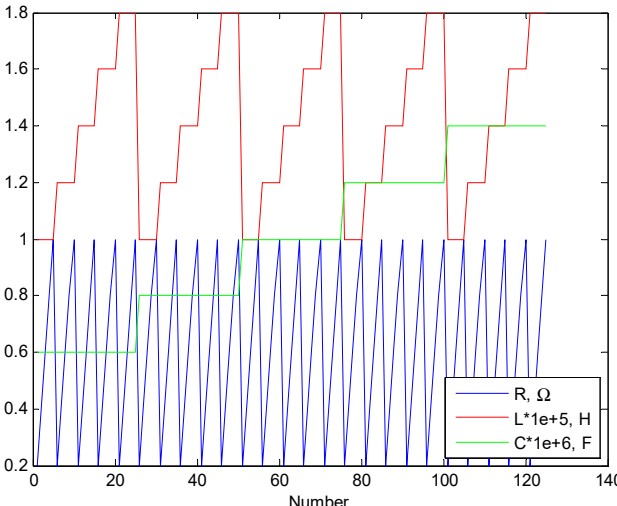

**Figure 5.** Generated data for $R$, $L$ and $C$ parameter values.

Then, using the mathematical model (from Figure 4), their corresponding data ($i_{aver}$, $\Delta i$, $u_{aver}$ and $\Delta u$) were calculated. The result of this calculation with the model is shown in Figure 6.

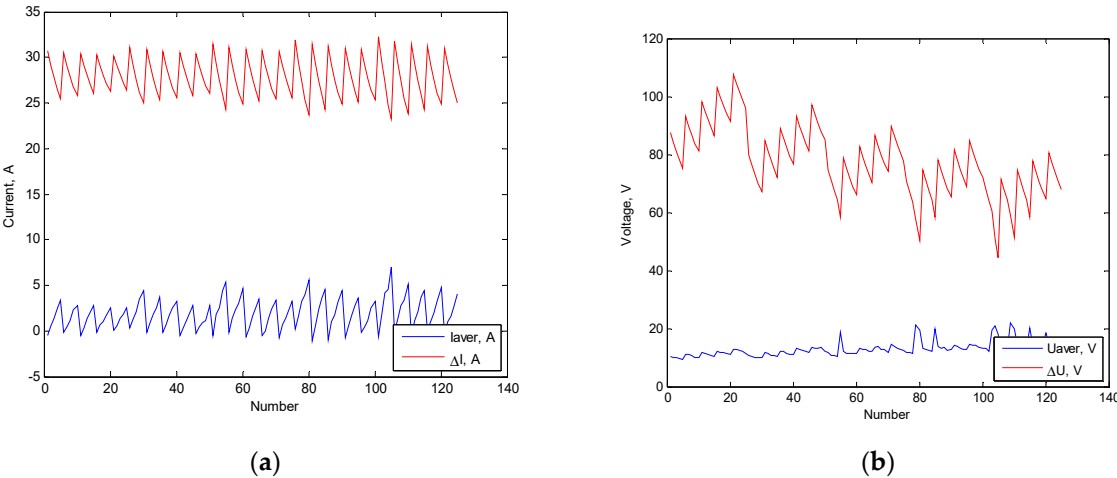

(**a**)                                        (**b**)

**Figure 6.** Output data for state variables: (**a**) current through the inductance $L$- $i_{aver}$ and $\Delta i$; (**b**) voltage across the capacitor $C$- $u_{aver}$ and $\Delta u$.

For a further solution to the design task, an NN-inverse model was created. At the time of input, the quantities $i_{aver}$, $\Delta i$, $u_{aver}$ and $\Delta u$ have averaged values, and at the output, the values of the circuit elements $R$, $L$ and $C$ are determined. Schematically, this model is shown in Figure 7. To train this network (the inverse network), we needed a large amount of diverse input and output data. With the data generated by the previous step of the automated design, the NN—inverse model was trained (shown in Figure 7).

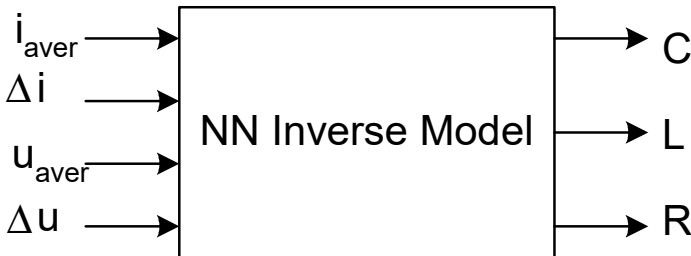

**Figure 7.** Inputs and outputs of neural networks.

Since neural networks are elements performing parallel calculations, the output values (the values of the elements of the single-ended DC/AC converter) are calculated all at once.

We chose a feed-forward neural network, the structure of which is shown in Figure 8. This network has one hidden layer and one output layer. We used "sigmoidal" activation function in the hidden layers, and the output layers are linear. It is well-known fact that a neural network with such a structure is a good approximator.

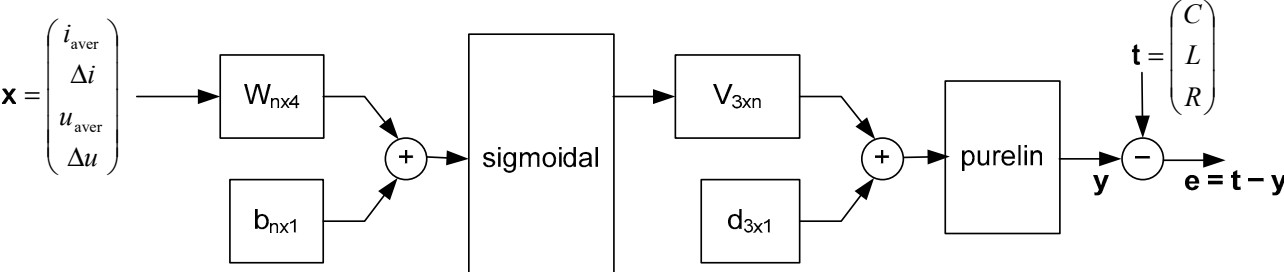

**Figure 8.** Structure of NN inverse model.

In the NN structure shown in Figure 8, only the number of neurons (n) in the first (hidden) layer is not specified. The number of these neurons determines the power of the network. The number (n) should be selected such that some of the unwanted cases (overlapping or underlapping) do not occur. In the process of training the network, a suitable value for the number was found to be $n = 60$. More details about the parameters and setup of the chosen neural network are given in Appendix A at the end of the manuscript.

Before starting the training of the NN-inverse model, we needed to rescale the data for the reference output $t = (C, L, R)^T$. This was necessary because the values for $R$ are several units, and the values for $L$ and $C$ are micro-H or -F. For this purpose, we multiplied $C$, $L$ and $R$ by $1 \times 10^6$, $1 \times 10^5$, and 1, respectively. The values for $C$, $L$ and $R$ are plotted precisely at this normalized scale in Figure 5.

The training of the neural network was based on the batch training method. Briefly, the method is as follows: Let the input vectors be the columns of the $X$ matrix and the outputs be the columns of the $Y$ matrix. For the selected neural network structure, we have:

$$Y = V \; sigmoidal( \; W \, X + B) + D \tag{6}$$

where the matrices $W$ and $B$ are the network weights in the first layer and the matrices $V$ and $D$ are the weights of the network in the second layer.

The mean squared error is, accordingly:

$$S(W, B, V, D) = \frac{1}{2}\sum_{k} ||t(k) - y(k)||^2 \tag{7}$$

where $k$ represents the time reports, $t(k)$ represents the reference network outputs, and $y(k)$ represents the realized outputs.

To complete the procedure, an optimization problem of the type is solved numerically:

$$\min_{(W, B, V, D)} S(W, B, V, D) \tag{8}$$

By default, the solution is by the Levenberg–Marquardt method. When solving problems of this type, the optimization procedure is considered complete when a certain minimum value of the objective function is reached or when a certain number of iterations are realized. Usually, an unattainably small value is chosen for the objective function, and in most practical cases, the optimization procedure terminates according to the "chosen maximum number of iterations" criterion. In the example shown, the iterations (called epochs in MATLAB) are set to 1000 by default.

We trained the neural networks in max1000 epochs (iteration), and mean squared errors, which are allowed in the training, are shown in Figure 9.

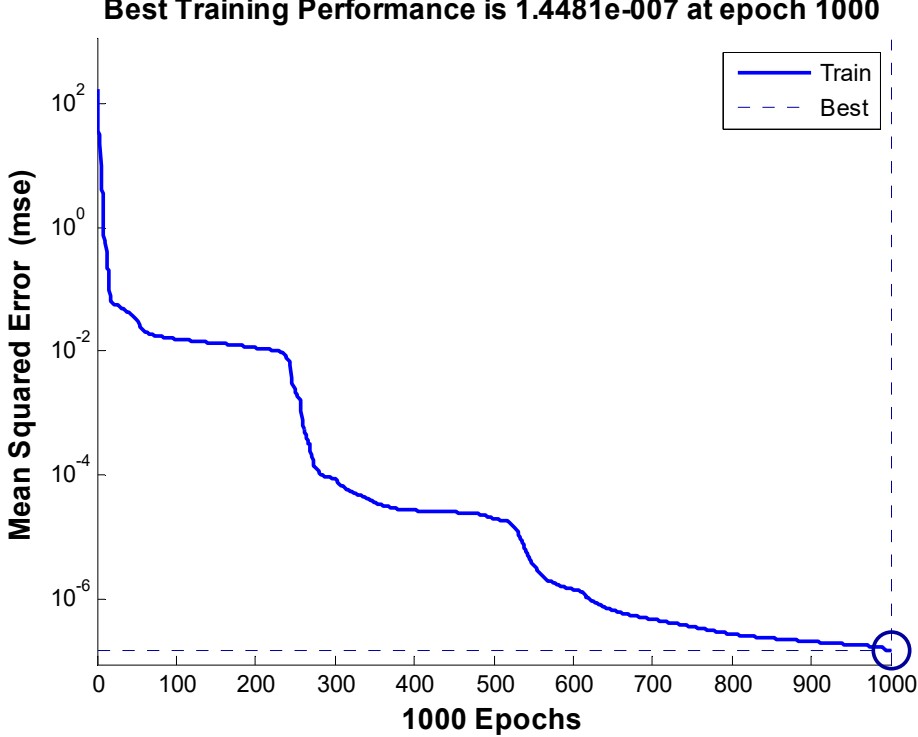

**Figure 9.** Mean squared error of NN training.

After the NN was already trained, the values for C, L, and R obtained by the neural network were compared with the values for C, L, and R that were used to generate $i_{aver}$, $\Delta i$, $u_{aver}$, and $\Delta u$ from the mathematical model. Then, $i_{aver}$, $\Delta i$, $u_{aver}$ and $\Delta u$ were fed as input to the neural network. In general, if the network is well-selected and well-trained, a good match between the *C*, *L* and *R* values given as input to the mathematical model and obtained as output of the neural network should be expected. The results given in Figure 10 show that such a match was achieved.

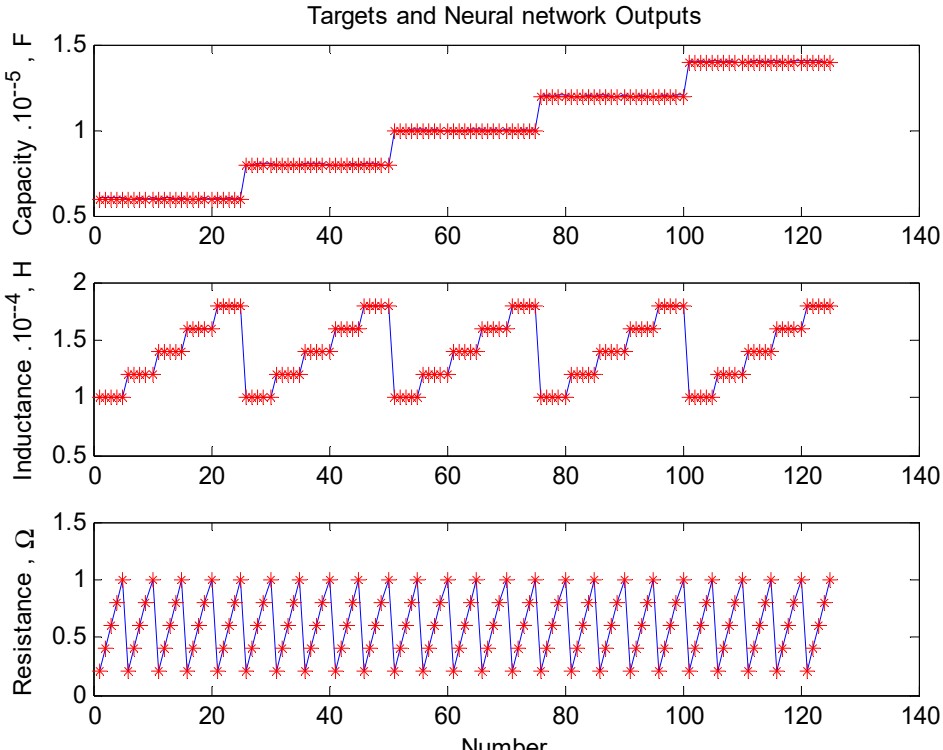

**Figure 10.** Targets (line) and neural network (stars) outputs.

### 3.3. Single-Ended DC/AC Converter Design Example

We verified the effectiveness of the design procedure using a specific example. For this purpose, we chose the following values for the inputs of the neural network:

$\Delta u$ = 85 V, $\Delta i$ = 27 A, $u_{aver}$ = 15 V and $i_{aver}$ = 2 A.

Using the neural network, the converter was designed and we obtained the following values for the circuit elements:

$C$ = 0.5132 µF, $L$ = 1.0781 µH and $R$ = 0.7619 Ω.

With the obtained values for $C$, $L$ and $R$, we performed computer simulations using the mathematical model of the DC/AC converter. The simulation results are shown in Figure 11.

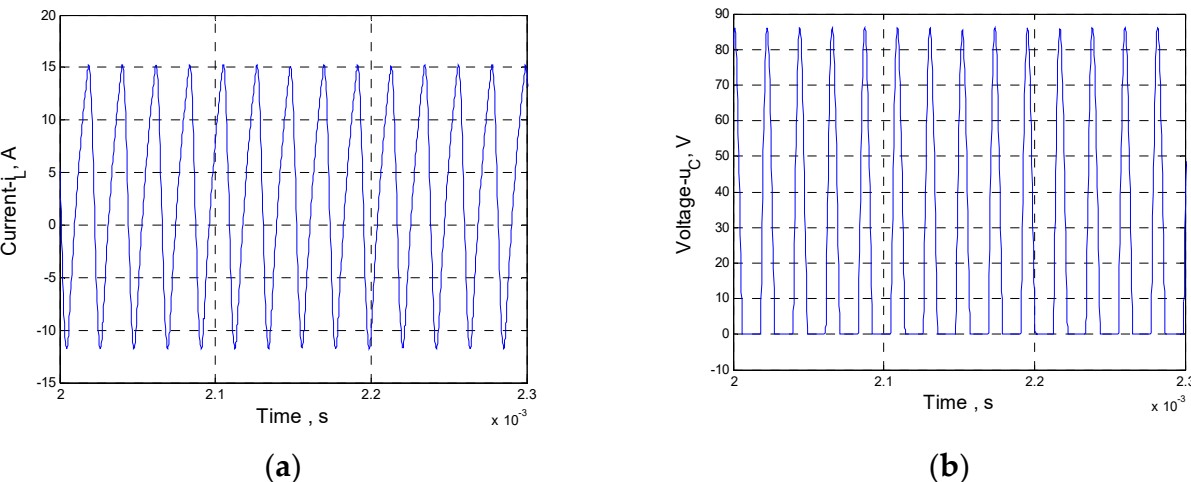

(**a**)          (**b**)

**Figure 11.** State variable results of the neural network-designed DC/AC converter: (**a**) current form through the inductance with parameters $i_{aver}$ and $\Delta i$; (**b**) voltage across the capacitor with parameters $u_{aver}$ and $\Delta u$.

For the data shown in Figure 11, the values of $i_{aver}$, $\Delta i$, $u_{aver}$ and $\Delta u$ were calculated. The calculation was performed with formulas (3) and (4). Of course, only the values of the state variables $i$ and $u$ in the established mode were used, as shown in Figure 11. The results were as follows:

$\Delta u$ = 86.0230 V, $\Delta i$ = 27.0437 A, $u_{aver}$ = 13.5835, and $i_{aver}$ = 1.8292 A.

We compared the obtained values of $i_{aver}$, $\Delta i$, $u_{aver}$ and $\Delta u$ with the values that were chosen for the input of the neural network at the beginning of this section. We found that they were extremely close (the difference was less than 5%). A certain (not large) deviation was observed only for the value of $u_{aver}$. However, this was most likely due to the fact that the values of $i_{aver}$, $\Delta i$, $u_{aver}$ and $\Delta u$ are mutually dependent and cannot be set arbitrarily (at the beginning, this was not taken into account when setting the values of the design parameters). These values were chosen according to the subjective desire of the designer. However, as a result of the analysis of the obtained results, it should be noted that an arbitrary combination between them cannot be achieved. Based on this, it should be concluded that the design of the DC/AC converter was successful and the set goal of applying neural networks for the automated design of power electronic devices was fully achieved.

### 4. Discussion and Conclusions

The paper presents a model-based and neural network-based innovative design of a single-ended transistor series resonant DC/AC converter with zero voltage switching (ZVS).

The applicability of this approach was proven through a specific computational example, and a satisfactory accuracy of 5% was achieved in the design, which is fully relevant to the requirements of the design of power electronic devices. By using mathematical modeling and implementing modern information and communication technologies, the design of the power circuit was automated. In this way, the dependence of the qualities of the designed device on the human factor was significantly reduced, thus formalizing and speeding up the process. The obtained results of the proposed method have an accuracy that is comparable to that obtained by classical design methods. The proposed method is indispensable for use in cases where there is no established methodology for the design of a given power electronic device, or where the calculation procedure is too complicated, without a clear physical interpretation, and/or based on different assumptions. The proposed innovative method does not negate classical design methods, but complements and develops them by applying artificial intelligence techniques. A natural extension of the research would be to perform an optimization procedure to further determine the values of the circuit elements using different objective functions. In this sense, it would be very useful to develop an interactive graphical user interface that would facilitate the use of the method by users who do not have specific knowledge of computer modeling and mathematical software. On the other hand, a disadvantage of the proposed design approach is the need to train the network. This could be compensated for by creating a database of designed devices, with subsequent additions and expansion occurring with each new implemented project.

It should be noted that in order to be effective, the method used must go through a process of normalization of the data, and it would be even more precise if the corresponding normalization were performed by the neural network itself, and not by the scientist or the user who created and uses it. On the other hand, the proposed approach is useful for the purposes of power electronics education because it enables the design of power electronic devices to be carried out even by those who do not have sufficient experience in or deep knowledge of this interdisciplinary field.

**Author Contributions:** N.H. and B.G. were involved in the full process of producing this paper, including conceptualization, methodology, modeling, validation, visualization, and preparing the manuscript. All authors have read and agreed to the published version of the manuscript.

**Funding:** This research was funded by the Bulgarian National Scientific Fund, grant number КП-06-Н57/7/16.11.2021, and the APC was funded by КП-06-Н57/7/16.11.2021.

**Institutional Review Board Statement:** Not applicable.

**Informed Consent Statement:** Not applicable.

**Data Availability Statement:** Not applicable.

**Acknowledgments:** This research was carried out within the framework of the projects: "Artificial Intelligence-Based modeling, design, control, and operation of power electronic devices and systems", КП-06-Н57/7/16.11.2021, Bulgarian National Scientific Fund.

**Conflicts of Interest:** The authors declare no conflict of interest.

## Appendix A

The appendix explains how to define the main parameters and settings of the neural network used in the manuscript.

The "feed forward" net, named net, was created with the following command:

net=newff(minmax(z),[60,3],{'logsig','purelin'});

In this command:

(a) minmax(z) is a standard command that finds the minimum and maximum values of each of the inputs, and this data is then used to initialize the weights of the two layers $W$, $B$, $V$, $D$;

(b) [60,3] shows that the network has two layers. The first (input) consists of 60 neurons (the number of neurons in this layer is not fixed in advance; this number determines the power of the network, and in the process of experimentation, a value of 60 neurons was selected) and the second (output) consists of 3 neurons (the number of neurons in the last layer is equal to the number of outputs, so there can only be 3 neurons here);

(c) {'logsig','purelin'} . . . show that the activation functions of the first and second layers are, respectively, logsig and purelin (subpoint "b" and "c" also determine the structure of the network shown in Figure 8).

The newff macro command generates, in the MATLAB Workspace, a net structure with the following fields, which contain the basic parameters of the neural network:

net =

Neural Network

name: 'Custom Neural Network'
efficiency: .cacheDelayedInputs, .flattenTime,
.memoryReduction
userdata: (your custom info)

dimensions:

numInputs: 1
numLayers: 2
numOutputs: 1
numInputDelays: 0
numLayerDelays: 0
numFeedbackDelays: 0
numWeightElements: 483
sampleTime: 1

connections:

biasConnect: [1; 1]
inputConnect: [1; 0]
layerConnect: [0 0; 1 0]
outputConnect: [0 1]

subobjects:

inputs: {1×1 cell array of 1 input}
layers: {2×1 cell array of 2 layers}
outputs: {1×2 cell array of 1 output}
biases: {2×1 cell array of 2 biases}
inputWeights: {2×1 cell array of 1 weight}
layerWeights: {2×2 cell array of 1 weight}

functions:

adaptFcn: 'adaptwb'
adaptParam: (none)
derivFcn: 'defaultderiv'
divideFcn: (none)
divideParam: (none)
divideMode: 'sample'
initFcn: 'initlay'
performFcn: 'mse'
performParam: .regularization, .normalization, .squaredWeighting
plotFcns: {'plotperform', plottrainstate,
plotregression}
plotParams: {1×3 cell array of 3 params}
trainFcn: 'trainlm'
trainParam: .showWindow, .showCommandLine, .show, .epochs,
.time, .goal, .min_grad, .max_fail, .mu, .mu_dec,
.mu_inc, .mu_max

weight and bias values:

IW: {2×1 cell} containing 1 input weight matrix
LW: {2×2 cell} containing 1 layer weight matrix
b: {2×1 cell} containing 2 bias vectors

methods:

adapt: Learn while in continuous use
configure: Configure inputs and outputs
gensim: Generate Simulink model
init: Initialize weights and biases
perform: Calculate performance
sim: Evaluate network outputs given inputs
train: Train network with examples
view: View diagram
unconfigure: Unconfigure inputs and outputs

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
