# Peer review of "Intelligent Design of ZVS Single-Ended DC/AC Converter Based on Neural Network"

_inventions, doi:10.3390/inventions8010041_

Round 1

Reviewer 1 Report

First of all, I would like to thank the authors for their valuable effort. However, there are some major comments that must be addressed.

1-     In my opinion, the introduction has some sort of ambiguity, each paragraph must be linked to the previous in a correlated way.

2-     Recently, most of the inverter control tends to accomplish grid code requirements. Therefore, I advise the authors to at least provide information regarding the possibility to introduce grid code requirements to the control system for the three-phase converter.

3-     A table with the parameters of the ANN must be introduced.

4-     The total number of cases to train the ANN must be mentioned.

5-     The article missing a comparison study with other papers in the field.

6-     In my opinion, the paper’s main contribution is unclear in the abstract in the conclusion. Please try to emphasize the main advantages of the paper, especially in the conclusion.

7-     The results represented in Fig. 11 do not show the exact evolution of the signal, therefore, it is recommended to make a zoom on one or two periods.

Author Response

First of all, we would like to thank you for your thorough review of our paper (inventions-2136380) and helpful comments to improve it.

Reviewer 1

Comments to the Authors
Comments and Suggestions for Authors

First of all, I would like to thank the authors for their valuable effort. However, there are some major comments that must be addressed.

1-     In my opinion, the introduction has some sort of ambiguity, each paragraph must be linked to the previous in a correlated way.

2-     Recently, most of the inverter control tends to accomplish grid code requirements. Therefore, I advise the authors to at least provide information regarding the possibility to introduce grid code requirements to the control system for the three-phase converter.

3-     A table with the parameters of the ANN must be introduced.

4-     The total number of cases to train the ANN must be mentioned.

5-     The article missing a comparison study with other papers in the field.

6-     In my opinion, the paper’s main contribution is unclear in the abstract in the conclusion. Please try to emphasize the main advantages of the paper, especially in the conclusion.

7-     The results represented in Fig. 11 do not show the exact evolution of the signal, therefore, it is recommended to make a zoom on one or two periods.

To Reviewer 1:

            Thank you very much for your review and valuable remarks.

  1. In my opinion, the introduction has some sort of ambiguity, each paragraph must be linked to the previous in a correlated way.

- Thank you very much for your comment. The entire manuscript has been substantially revised. 

  1. Recently, most of the inverter control tends to accomplish grid code requirements. Therefore, I advise the authors to at least provide information regarding the possibility to introduce grid code requirements to the control system for the three-phase converter.

- Thank you very much for your comment. Control synthesis of the DC/AC converter is not the subject of the current research, also the considered converter is not intended for grid operation. 

  1. A table with the parameters of the ANN must be introduced..

- Thank you very much for the recommendation and suggestion. An appendix has been added at the end of the manuscript describing the parameters and settings of the neural network used.

  1. The total number of cases to train the ANN must be mentioned.

-  Thank you very much for the recommendation and suggestion. Added detailed explanation in the text: When solving problems of this type, the optimization procedure is considered complete when a certain minimum value of the objective function is reached or until a certain number of iterations are realized. Usually, an unattainably small value of the objective function is chosen, and in most practical cases the optimization procedure terminates according to the "chosen maximum number of iterations" criterion. In the example shown, the iterations (called epochs in MATLAB) are set to 1000 by default.

  1. The article missing a comparison study with other papers in the field.

- Thank you very much for your comment. Such researches on the application of neural networks in the design of power electronic devices are lacking, rather there are those on choosing the appropriate topology, by applying artificial intelligence techniques, and therefore the title of the manuscript is "An Innovative Design Approach..."

  1. In my opinion, the paper’s main contribution is unclear in the abstract in the conclusion. Please try to emphasize the main advantages of the paper, especially in the conclusion.

- Thank you very much for your comment. The manuscript has been substantially revised, with relevant clarifications included in the introduction and conclusion sections. 

7-     The results represented in Fig. 11 do not show the exact evolution of the signal, therefore, it is recommended to make a zoom on one or two periods.

- Thank you very much for the recommendation. The figure has been edited. 

 Thank you very much for your remarks and comments. They were very useful for me to emphasize the main tasks and contributions of the manuscript, and also to focus the readers attention on the new and unique elements.

Reviewer 2 Report

Most bibliographical references, except [26, 27, and 28], have not been cited in the manuscript's text. All references must be cited in the text.

The editing of the text of the manuscript (position of the figures, different font sizes of the equations) should be improved, to optimize the size of the manuscript.

The following spelling errors have been detected and should be corrected:

Page 1, line 11. “Characteristic of the proposed design …”, (A characteristic of …)

Page 1, line 36. “This is due both to the development …”, (both should be eliminated in this paragraph).

Page 1, line 40. “… fields of research in the last few decades…”, (it is more suitable “research fields”)

Page 2, line 85. “In [26, 27], the authors applied an approach for the design of …” (it is better to say to design).

Page 2, line 88. “… since the verification of the obtained results is easy.” (since verifying the …”)

Page 3, line 100. “… The scheme consists of: load with an active-inductive …” (a load).

Page 3, line 123. “… the capacitor C is charged according to …” (article the should be eliminated)

Page 4, line 150. “Using Kirchhov's laws, …” (Kirchhoff’s laws)

Page 5, lines 154-155. “… It therefore operates …” (, therefore,)

Page 5, line 162. “In Figure 3 shows …” (eliminate “in”)

Page 6, line 179. “… are calculated some time after the start …” (sometimes)

Page 6, line 180. “… in this case the interval is …” (case,)

Page 6, line 204. “… first, that the condition for resonance of the series circuit is satisfied …” (eliminate “that”), (include for “the” resonance)

Page 8, line 235. “… with structure shown on …” (… the structure shown in …)

Page 8, lines 252-253. “… and the outputs be the columns of …” (are)

Page 8, line 264. “… are allowed on the train is shown on Figure 9.” (in)

Page 9, lines 272-273. “… The results given in Figure 10 show that …” (shows)

Page 10, line 312. “… The use of the proposed method is …” (it is more adequate to eliminate “use of the”).

Page 11, lines 331 and 332. Eliminate “Authorship must be limited to those who have contributed substantially to the work reported.”

Author Response

First of all, we would like to thank you for your thorough review of our paper (inventions-2136380) and helpful comments to improve it.

Reviewer 2

Comments to the Authors

Most bibliographical references, except [26, 27, and 28], have not been cited in the manuscript's text. All references must be cited in the text.

The editing of the text of the manuscript (position of the figures, different font sizes of the equations) should be improved, to optimize the size of the manuscript.

The following spelling errors have been detected and should be corrected:

Page 1, line 11. “Characteristic of the proposed design …”, (A characteristic of …)

Page 1, line 36. “This is due both to the development …”, (both should be eliminated in this paragraph).

Page 1, line 40. “… fields of research in the last few decades…”, (it is more suitable “research fields”)

Page 2, line 85. “In [26, 27], the authors applied an approach for the design of …” (it is better to say to design).

Page 2, line 88. “… since the verification of the obtained results is easy.” (since verifying the …”)

Page 3, line 100. “… The scheme consists of: load with an active-inductive …” (a load).

Page 3, line 123. “… the capacitor C is charged according to …” (article the should be eliminated)

Page 4, line 150. “Using Kirchhov's laws, …” (Kirchhoff’s laws)

Page 5, lines 154-155. “… It therefore operates …” (, therefore,)

Page 5, line 162. “In Figure 3 shows …” (eliminate “in”)

Page 6, line 179. “… are calculated some time after the start …” (sometimes)

Page 6, line 180. “… in this case the interval is …” (case,)

Page 6, line 204. “… first, that the condition for resonance of the series circuit is satisfied …” (eliminate “that”), (include for “the” resonance)

Page 8, line 235. “… with structure shown on …” (… the structure shown in …)

Page 8, lines 252-253. “… and the outputs be the columns of …” (are)

Page 8, line 264. “… are allowed on the train is shown on Figure 9.” (in)

Page 9, lines 272-273. “… The results given in Figure 10 show that …” (shows)

Page 10, line 312. “… The use of the proposed method is …” (it is more adequate to eliminate “use of the”).

Page 11, lines 331 and 332. Eliminate “Authorship must be limited to those who have contributed substantially to the work reported.”

To Reviewer 2:

            Thank you for your review and valuable remarks.

  1. Most bibliographical references, except [26, 27, and 28], have not been cited in the manuscript's text. All references must be cited in the text.

- Thank you very much for your comment. All literary sources are cited in the text.

  1. The editing of the text of the manuscript (position of the figures, different font sizes of the equations) should be improved, to optimize the size of the manuscript.

- Thank you very much for your comment. All equations have been reviewed and corrected.

  1. The following spelling errors have been detected and should be corrected:

Page 1, line 11. “Characteristic of the proposed design …”, (A characteristic of …)

Page 1, line 36. “This is due both to the development …”, (both should be eliminated in this paragraph).

Page 1, line 40. “… fields of research in the last few decades…”, (it is more suitable “research fields”)

Page 2, line 85. “In [26, 27], the authors applied an approach for the design of …” (it is better to say to design).

Page 2, line 88. “… since the verification of the obtained results is easy.” (since verifying the …”)

Page 3, line 100. “… The scheme consists of: load with an active-inductive …” (a load).

Page 3, line 123. “… the capacitor C is charged according to …” (article the should be eliminated)

Page 4, line 150. “Using Kirchhov's laws, …” (Kirchhoff’s laws)

Page 5, lines 154-155. “… It therefore operates …” (, therefore,)

Page 5, line 162. “In Figure 3 shows …” (eliminate “in”)

Page 6, line 179. “… are calculated some time after the start …” (sometimes)

Page 6, line 180. “… in this case the interval is …” (case,)

Page 6, line 204. “… first, that the condition for resonance of the series circuit is satisfied …” (eliminate “that”), (include for “the” resonance)

Page 8, line 235. “… with structure shown on …” (… the structure shown in …)

Page 8, lines 252-253. “… and the outputs be the columns of …” (are)

Page 8, line 264. “… are allowed on the train is shown on Figure 9.” (in)

Page 9, lines 272-273. “… The results given in Figure 10 show that …” (shows)

Page 10, line 312. “… The use of the proposed method is …” (it is more adequate to eliminate “use of the”).

Page 11, lines 331 and 332. Eliminate “Authorship must be limited to those who have contributed substantially to the work reported.”

- Thank you very much. Necessary corrections have been made. I have inadvertently uploaded a version of the manuscript before stylistic correction.

 Thank you very much for your remarks and comments. They were very useful for me to emphasize the main tasks and contributions of the manuscript, and also to focus the readers attention on the new and unique elements.

Reviewer 3 Report

1. The representations for NN should add into the equation's numbers from page 254 to 261.

2.  The values of parameters W, V, b,d in NN should be listed.

3. The units of x-axis in Figures 5, 6, 10 should be listed.

4. The values in Figure 11 should be amplified.

5. The parameters  R,L,C should be added into units.   

Author Response

First of all, we would like to thank you for your thorough review of our paper (inventions-2136380) and helpful comments to improve it.

Reviewer 3

Comments to the Authors

Comments and Suggestions for Authors

  1. The representations for NN should add into the equation's numbers from page 254 to 261.
  2. The values of parameters W, V, b,d in NN should be listed.
  3. The units of x-axis in Figures 5, 6, 10 should be listed.
  4. The values in Figure 11 should be amplified.
  5. The parameters R,L,C should be added into units.

To Reviewer 3:

            Thank you for your review and valuable remarks.

  1. The representations for NN should add into the equation's numbers from page 254 to 261.

- Thank you very much for the recommendation and suggestion. An appendix has been added at the end of the manuscript describing the parameters and settings of the neural network used.

  1. The values of parameters W, V, b, d in NN should be listed.

- Thank you very much for the remark. The meaning of these parameters is described in the text.

  1. The units of x-axis in Figures 5, 6, 10 should be listed.

- Thank you very much for the remark. The figures have been corrected.

  1. The values in Figure 11 should be amplified.

- Thank you very much for the remark. The figure has been corrected.

  1. The parameters R,L,C should be added into units.

- Thank you very much for the remark. The units of measure of schematic elements have been added.

 Thank you very much for your remarks and comments. They were very useful for me to emphasize the main tasks and contributions of the manuscript, and also to focus the readers attention on the new and unique elements.

Round 2

Reviewer 1 Report

thank you, all my comments have been addressed

Reviewer 3 Report

This paper has been revised according to the reviewer's comments. This paper should be accepted.